# Mechanism of Heterogeneous Alkaline Deacetylation of Chitin: A Review

**DOI:** 10.3390/polym15071729

**Published:** 2023-03-30

**Authors:** Vitaly Yu. Novikov, Svetlana R. Derkach, Irina N. Konovalova, Natalya V. Dolgopyatova, Yulya A. Kuchina

**Affiliations:** 1Polar Branch of Russian Federal Research Institute of Fisheries and Oceanography, Murmansk 183038, Russia; 2Institute of Natural Science and Technology, Murmansk State Technical University, Murmansk 183010, Russia

**Keywords:** chitin, chitosan, deacetylation, kinetics, reaction mechanism, hydration

## Abstract

This review provides an analysis of experimental results on the study of alkaline heterogeneous deacetylation of chitin obtained by the authors and also published in the literature. A detailed analysis of the reaction kinetics was carried out considering the influence of numerous factors: reaction reversibility, crystallinity and porosity of chitin, changes in chitin morphology during washing, alkali concentration, diffusion of hydroxide ions, and hydration of reacting particles. A mechanism for the chitin deacetylation reaction is proposed, taking into account its kinetic features in which the decisive role is assigned to the effects of hydration. It has been shown that the rate of chitin deacetylation increases with a decrease in the degree of hydration of hydroxide ions in a concentrated alkali solution. When the alkali concentration is less than the limit of complete hydration, the reaction practically does not occur. Hypotheses have been put forward to explain the decrease in the rate of the reaction in the second flat portion of the kinetic curve. The first hypothesis is the formation of “free” water, leading to the hydration of chitin molecules and a decrease in the reaction rate. The second hypothesis postulates the formation of a stable amide anion of chitosan, which prevents the nucleophilic attack of the chitin macromolecule by hydroxide ions.

## 1. Introduction

Chitosan is a deacetylated derivative of chitin, the second most common natural polymer after cellulose. Chitin is the main component of the exoskeletons of crustaceans, such as crabs, lobsters, shrimps [1,2,3], and insects, the radula of mollusks, the scales of fish and lissamphibians, and the cell walls of fungi [4,5]. In chitinous organisms, chitin is found in complexes with proteins and glucans.

Chitin has three polymorphic modifications called alpha-, beta- and gamma-chitins with different orientations of microfibrils [6]. The reason for the difference between these polymorphs lies in the biosynthesis of chitin by different organisms to perform different functions. The most common is the α-form, present in virtually all organisms including crustaceans and insects; it is densely packed antiparallel polymer chains. β-chitin is found in mollusks (squid, cuttlefish). In the case of the β-form, the polymer chains are parallel and, due to weaker intermolecular hydrogen bonds, have greater solubility and swelling capacity [7]. In microorganisms, a mixed form is found, called γ-chitin [8].

Chitin is insoluble in water, alkalis, dilute acids, alcohols, and other organic solvents [4,9]. It is able to form complexes with proteins, peptides, and cholesterol. Additionally, it has a high sorption capacity for heavy metals and radionuclides [10]. Among the various derivatives of this biopolymer, chitosan is the most accessible.

Chitin is a polymeric carbohydrate, consisting of a linear structure of β-(1→4)-N-acetyl-D-glucosamine main monomer (Figure 1a), and chitosan is a polymeric carbohydrate consisting of a linear structure of β-(1→4)-D-glucosamine main monomer (Figure 1b). Strictly speaking, though, chitin and chitosan are linear random copolymers of (1→4)-linked-2-acetamido-2-deoxy-β-D-glucan (acetylated units) and (1→4)-linked-2-amino-2-deoxy-β-D-glucan (deacetylated units) (Figure 1c).

Chitin and chitosan are distinguished by the content of amino groups (or deacetylated units). However, there are currently no generally accepted quantitative criteria. In some cases, this conditional boundary can be drawn according to the degree of deacetylation (DD), which is less than 50% for chitin and more than 50% for chitosan [11]. So, for chitin extracted from natural raw materials, DD ranges from 5 to 15%. Chitosan with DD ≥ 50% is soluble in an aqueous acidic environment [12,13]. The degree of deacetylation (DD, in molar percentage) is the ratio of the number of glucosamine units (acetylated units) to the total number of monomer units (acetylated, n, and deacetylated, m, units) in a polysaccharide molecule:DD (%) = 100 × n/(n + m),(1)

Chitosan, unlike practically insoluble chitin, is soluble in dilute inorganic and some organic (formic, acetic, succinic, lactic, malic) acids [14,15]. The amino groups of the chitosan molecule have an ionic dissociation constant (pKa) of 6.3–6.5 [16]. At pH values below this, the amino groups are protonated, and chitosan is a cationic, highly soluble polyelectrolyte. At higher pH values, the polymer is insoluble. This dependence of solubility on pH allows for the use of chitosan in various forms: solutions, suspensions, nanoparticles [17], bionanocomposites [18], gels [19,20,21,22], capsules [23], films, membranes [24], fibers [25], etc. The solubility of chitosan in slightly acidic aqueous solutions increases significantly with a decrease in molecular weight and an increase in the degree of deacetylation [26].

The presence of an additional functional group (amino group) makes it possible to obtain N-derivatives of various types from chitosan [1], which significantly expands the scope of its application. Due to the presence of a positive charge, it has an affinity for the sorption of molecules of proteins [13], pesticides, dyes, and lipids, and the chelation of metal ions and radionuclides [27,28]. Products based on chitosan are biodegradable and biocompatible. Chitosan and its derivatives are non-toxic and have biological activity. They exhibit antibacterial, immunostimulating, antitumor, wound healing, and other properties [4,29]. The unique properties of this biopolymer lead to its increased use in many fields: the food industry and nutrition [26], medicine [29,30,31] and biomedicine [20,32,33], pharmacy [13], cosmetology [11], agriculture [34,35], paper industries and functional textiles [36,37], the edible film industry and packaging [38], environmental chemistry [39], biotechnology [40], nanotechnology [41], and aquaculture [4,26,42].

The production of chitosan is based on the reaction of chitin deacetylation (the reaction of hydrolysis of acetamide groups) [43], which is accompanied by the cleavage of the acetyl group from the structural unit of chitin with the formation of an amino group (Figure 2). The deacetylation reaction can be accompanied by a simultaneous rupture of glycosidic bonds of the polymer and, accordingly, a decrease in the molar mass, therefore, in many cases, chitosan has a structural heterogeneity due to the rupture of the polymer chain [44].

Chitosan can be obtained from chitin chemically [1,45] or by using enzymatic preparations [2,46]. From a chemical point of view, both acids and alkalis can be used to deacetylate chitin. However, alkaline deacetylation is used more often [3,45] since glycosidic bonds are very sensitive to an acidic environment, in which they are destroyed [47].

Chitin deacetylation in an alkaline medium occurs either in a heterogeneous [7] or homogeneous medium [48]. Usually, according to the heterogeneous method, chitin is treated with a concentrated NaOH solution at a high temperature for several hours. As a result, chitosan is formed as an insoluble residue with a degree of deacetylation up to ~80–90%. According to the homogeneous method, chitin is dispersed in a concentrated NaOH solution at room temperature for several hours, followed by dissolution at ~0 °C. The degree of deacetylation of the resulting chitosan can reach 90% [49]. Therefore, it seems difficult to produce high quality chitosan with DD greater than 90% without chain degradation (molecular weight reduction). Numerous results show that the properties of chitosan, the degree of deacetylation, and molecular weight distribution significantly depend on the reaction time, alkali concentration, temperature, and the repetition of alkaline steps [50].

The chemical method of deacetylation has its drawbacks: energy consumption, concentrated alkaline solution waste, and other chemical wastes that require disposal to prevent environmental pollution. To overcome these shortcomings, an alternative enzymatic method using chitin deacetylase [1] and other eco-friendly extraction methods [51] were proposed. Thus, enzymatic hydrolysis using preparations based on chitin deacetylases makes it possible to control the process leading to the production of chitosan oligomers [52]. However, the disadvantages of this method are lower quality and higher cost compared to the chemical method. Detailed information about the methods for obtaining chitosan is presented in numerous excellent reviews and monographs such as, for example, [4,14,50].

Chemical methods for producing chitosan are widely used for industrial purposes due to their low cost and suitability for mass production [53]. Currently, the industrial method for producing chitosan is based on the reaction of chitin deacetylation using a concentrated alkali solution (NaOH or KOH) proceeding at high temperatures (90–120 °C) under heterogeneous conditions [4].

Nevertheless, an important issue still remains unresolved. It is associated with a sharp decrease in the rate of the chitin deacetylation reaction after 60–90 min from the start of the process when DD reaches 65–75%. A further increase in DD to 85–90% occurs under conditions of a low reaction rate for several hours. This has been commonly reported in the literature [54,55,56,57]. In practice, the degree of deacetylation of 100% under a single treatment with alkali is unattainable [7,58]. A high DD is achieved by multiple deacetylation processes [59,60].

All of the main unique properties of the chitosan polysaccharide depend mainly on the degree of deacetylation (and on the molecular weight) [49,61]. Therefore, studies to understand the nature of the behavior of the chitin deacetylation reaction and to find ways to quickly increase the degree of deacetylation of the final product, chitosan, up to 100%, are relevant and in demand.

In this review, based on the analysis of the literature’s data and the authors’ own work, we propose a mechanism for the alkaline deacetylation of chitin under heterogeneous conditions, which explains the features of the deacetylation reaction. Establishing the mechanism is of practical importance for improving the industrial production of chitosan and makes a theoretical contribution to the development of ideas about the effect of the solvent and hydration processes on the kinetics of alkaline heterogeneous deacetylation of chitin.

The review consists of an introduction, a conclusion, and three parts. In the first part, the kinetic regularities of the chitin deacetylation reaction are considered. In the second part, the influence of various factors on the reaction kinetics of deacetylation is considered. The third part is devoted to a discussion of the proposed mechanism of the reaction occurring in an alkaline medium under heterogeneous conditions, which explains the discovered kinetic patterns.

## 2. Kinetics of the Chitin Deacetylation Reaction

Most often, the kinetic regularities of the chitin deacetylation reaction, the product of which is chitosan, are studied using the example of chitin obtained from different types of marine crustaceans [1]. The variety of raw sources of chitin led to a certain scatter and incompatibility of the results, which required the systematization of the accumulated material in order to identify possible differences between chitin samples from different organisms.

We compared the kinetic curves (degree of deacetylation versus time) and physicochemical properties of chitin isolated from the shells of these marine crustaceans. Data published in the scientific literature [3,7] and our own experimental data [44,47,62,63] show that α-chitin obtained from various raw materials (crab shells, shrimp, Antarctic krill, as well as the internal organs of crustaceans, for example, the gills of the king crab [64]), is almost identical both in its chemical composition and in physicochemical properties. A similar conclusion was made in [65].

Kinetic curves of chitin deacetylation from the shells of three different crustaceans and the gills of the king crab are shown in Figure 3. These curves coincide within the realm of experimental error. The kinetic curve has a characteristic shape with an initial section with a sharp increase in the degree of deacetylation over about 30 min (first section, fast deacetylation section) followed by a section of slow increase in DD over time (second section, slow deacetylation section). After 3–4 h of the reaction, the conversion limit is reached, which is about DD ≈ 80%. Carrying out the reaction for a longer time does not lead to a significant increase in the degree of deacetylation and does not make it possible to obtain chitosan with DD = 100%. This has been repeatedly confirmed by data published in the literature [57,66,67,68] and various models have been proposed to describe the kinetics of deacetylation.

Generally, chitin deacetylation can be considered as a non-catalytic liquid-solid reaction. The kinetics can be described by the shrinking core model [68], which is applicable for the initial time interval (0–30 min) of the reaction. A modification of this model [69] takes into account the diffusion capacity of the reagent inside the polymer and makes it possible to predict the degree of deacetylation over the entire time interval of the process, taking into account the unattainability of a degree of deacetylation of 100%.

We have shown that the kinetic curves of chitin/chitosan deacetylation with different initial degrees of deacetylation (DD_0_) from 15 to 95% (Figure 4, curves 1–4) can be transformed into a “standard” kinetic curve (Figure 4, curve 5) built within the coordinates of the reduced degree of deacetylation (DDtotal) versus time. This indicates the fulfilment of the condition of Invariant I [70]:DD_total_ (%) = (DD − DD_0_)/(100 − DD_0_),(2)
where DD_total_ is the reduced degree of deacetylation, DD is the degree of deacetylation at a given time, and DD_0_ is the initial degree of deacetylation.

The data obtained (Figure 4) indicate that the kinetics of the deacetylation reaction do not depend on the initial degree of deacetylation of the polysaccharide. The fulfilment of the condition of Invariant I (see Equation (2)) shows that the reaction mechanism includes a system of reactions of the first or pseudo-first order throughout the entire deacetylation process [71]. This is primarily due to the excess of one of the reactants of the reaction, namely, alkali. In the literature, the deacetylation of the acetamide groups is considered to be a first-order process [43,58,65,72]. The slow deacetylation rate is the same order as those reported for the deacetylation of chitin derived from mud crab shells [65], shrimp shells [56], and squid [73], and is in the range (1.6–5.9) × 10^−3^ min^−1^. The estimated activation energy [59,60,65] for heterogeneous deacetylation of chitin derived from different crustacean shells is in the range (16.2–56.0 kJ mol^−1^).

### 2.1. Factors Determining the Inhibition of the Deacetylation Reaction

To explain the features of the kinetics of the chitin deacetylation reaction—namely, the presence of fast and slow deacetylation regions on the kinetic curve, various hypotheses were proposed: the existence of a reverse acetylation reaction [15], the presence of crystalline and amorphous regions in chitin [74], changes in the morphology of partially deacetylated chitin during washing, the decrease in alkali concentration during deacetylation [75], diffusion of hydroxide ions and reaction products and the effect of chitin particle porosity [76,77,78,79], the existence of two parallel reactions leading to the formation of different products, the formation of a reaction inhibitor or inactive intermediates (complexes, hydrates), and the effect of hydration (solvation) of reacting particles on their interaction with chitin [15,44].

The reaction system of chitin–aqueous concentrated solution of NaOH (KOH) contains a limited number of reagents: an insoluble porous substance (chitin), water, and dissolved alkali in the form of hydrated OH^−^ and Na^+^ (or K^+^) ions, which makes the system convenient for research. After the reaction, a porous substance, chitosan, and sodium acetate are formed, which in solution gives a hydrated acetate anion CH_3_COO^−^ and a hydrated alkali metal cation. These substances determine the set of existing reactions that can take place during deacetylation. Let us separately consider the possible influence of all the factors listed above on the deacetylation process.

### 2.2. Reversibility of the Reaction

The observed shape of the deacetylation kinetic curve (Figure 3 and Figure 4) immediately leads to the assumption of the existence of a reversible reaction and the establishment of an equilibrium state in the reaction mixture (see Figure 2). However, as is known from the chemical properties of organic amines, the conditions under which deacetylation is carried out exclude the possibility of the reaction of acetylation of the amino group with sodium acetate.

Our studies have shown that when an excess of sodium acetate is added to the reaction mixture, the reverse acetylation reaction, which shifts the equilibrium towards the formation of chitin, is not observed (Figure 5). It was also found that the concentration of NaOH during deacetylation decreases insignificantly from 50 to 49.4%, and the addition of an excess of NaOH to the reaction mixture does not cause an increase in DD in the second section of the kinetic curve.

### 2.3. Crystallinity of Chitin

It has been suggested that the presence on the kinetic curve (Figure 3) of areas of fast and slow deacetylation is explained by the presence of amorphous and crystalline [80] regions in chitin. This hypothesis was put forward in [74] and continues to be used in various works [59,65,72]. It is assumed that in the amorphous regions, the deacetylation reaction proceeds rapidly (the first section of the kinetic curve), while in the crystalline regions, the reaction rate is lower due to the slow diffusion of the reacting particles (alkali molecules or hydroxide ions) to the chitin molecules. Therefore, after the complete deacetylation of amorphous regions, the deacetylation rate decreases, and the second section of the kinetic curve appears.

For example, in [81] it was shown that the deacetylation of the chitin occurred in two stages, which were respectively attributed to the reaction of acetamide groups in the amorphous region on the external layer, and in the crystalline region inside the chitin particles. The activation energy of the slow deacetylation stage was estimated to be 32.6 kJ mol^−1^ [65].

At the same time, experimental material has been accumulated that raises doubts about the correctness of the hypothesis about the exclusive effect of chitin crystallinity on the kinetics of the deacetylation reaction. Thus, it was shown in [82] that with an increase in the degree of deacetylation, the degree of crystallinity (χ_cr_) of chitin decreases, and then (at DD above 90–95%) it begins to increase again as a result of the formation of a new crystal structure of chitosan. Deacetylation can be accompanied by a monotonic decrease in the crystallinity of chitin [83]. At the same time, our results confirmed the implementation of the transformation of Invariant I (see Equation (2)) for the kinetic curves of chitin/chitosan deacetylation in almost the entire range of initial values of DD0 from 15.6 to 93% (Figure 4), i.e., the absence of influence of the crystallinity of the initial polysaccharide to the deacetylation process.

Confirmation of the absence of the effect of crystallinity on the kinetics was obtained [62] by studying the reaction of deacetylation of chitin samples with different degrees of crystallinity (χ_cr_ = 61.1 and 8.5%) and humidity (Figure 6). The X-ray scattering diffraction patterns of chitin shown in Figure 7 are in accordance with those in the literature [84]. In the first section of the kinetic curve, the highest rate of the deacetylation reaction is observed in the initial dry sample of chitin with the highest degree of crystallinity χ_cr_ = 61.1% (Figure 6, curve 1). With a decrease in the degree of crystallinity (χ_cr_ = 60.7%, dry chitin reprecipitated from HCl solution), the initial rate of deacetylation decreases (Figure 6, curve 3). The lowest reaction rate is observed in a wet chitin sample reprecipitated from a hydrochloric acid solution with the lowest degree of crystallinity of the available samples χ_cr_ = 8.5% (Figure 6, curve 4).

The influence of the degree of crystallinity of chitin and chitosan on the rate of the deacetylation reaction cannot be completely ruled out, but this factor definitely does not make a significant contribution to the inhibition of the reaction in the second section of the kinetic curve.

### 2.4. Diffusion of Alkali

The results of studies of the kinetics of the deacetylation reaction obtained by ^1^H and ^13^C NMR spectroscopy [76,77], indicate that the main role is played by the rate of diffusion of alkali into the solid particles of chitin. Confirmed by data in the literature [56,85], the observed decrease in the deacetylation rate is mainly ascribed to diffusion limitations. It is assumed that the rate of heterogeneous deacetylation can be controlled by the effective diffusivity of OH^−^ through the reacted (deacetylated) layer inside the polysaccharide particles [68]. According to the model proposed in [69], the remarkable decrease of the deacetylation rate is attributed to a strong reduction in the diffusivity of the reactant within the polymer preventing the degree of deacetylation reaching 100%, even in the presence of excess NaOH.

The effect of diffusion will apparently be observed in samples of recrystallized chitin, which is characterized by the close packing of molecules, as well as in samples of finely ground chitin with a disturbed crystal structure [86,87].

The influence of the diffusion of reagents on the rate of the deacetylation reaction was confirmed in [78,79]. The effect of the preliminary evacuation of the reaction mixture on the rate of deacetylation of α- and β-chitin was shown. A sharp increase in the deacetylation rate in the first region was observed upon preliminary deaeration of chitin, which is explained by a sharp increase in the rate of diffusion of alkali into the pores of chitin after preliminary evacuation.

The reason for stopping the reaction at the site of slow deacetylation seems to be associated with the formation of acetate ions which fill the pores and prevent further diffusion of alkali molecules into deeper regions of chitin. After washing the deacetylation products (sodium acetate), the chitin pores are released, which facilitates the diffusion of NaOH molecules into pores of chitin particles. A similar conclusion was made in [68,88]. In this regard, it is necessary to take into account the effect of chitin porosity on the deacetylation kinetics.

### 2.5. Porosity of Chitin

The porosity of chitin is due to the structure of the shell of crustaceans [89]. In order to establish the effect of the porosity of chitin particles on the kinetics of deacetylation, we studied the deacetylation of chitin with different supramolecular structures—the pore volume, specific surface area, pore diameter, and crystallinity [53,62]. When dried in air with a gradual evaporation of water, flexible chitin fibrils approached each other, which led to the disappearance of small pores. During freeze-drying, the primary structure of chitin is preserved, therefore, chitin samples dried differently differ in pore size.

In freeze-dried chitin (Figure 8, sample 2), the specific surface area and pore volume are higher compared to air-dried (Figure 8, sample 1). When reprecipitated chitin was dried in air (sample 3), its porosity (pore size) turned out to be minimal due to the denser packing of chitin molecules. The porosity of chitin after freeze-drying (sample 4) is close to the porosity of the original samples.

Chitin samples with different pore sizes (Figure 8) deacetylate almost identically (Figure 9). However, the low porosity of the reprecipitated air-dried chitin (Figure 8, sample 3) leads to the rate of deacetylation in the first section (Figure 9, curve 3) as a result of the slow wetting of the interior of the polysaccharide particles. Long-term deacetylation of all chitin samples resulted in classical kinetic curves that do not reach 100% DD. After a six-hour treatment of the original and reprecipitated chitin samples, DD was 74 and 82%, respectively. The model proposed in [66] integrates both kinetics and transport phenomena (transport resistance for OH^−^) and incorporates the data for particle morphology and porosity, which ensured a relatively good agreement for the experimental data by considering the kinetics.

An analysis of the results allows us to state that porosity and crystallinity affect the deacetylation kinetics but do not change the overall picture of the observed deceleration of the deacetylation rate. The existence of almost the same “plateau” on the kinetic curves after 20–30 min from the start of the reaction indicates not a diffusion, but rather the kinetic nature of the decrease in the deacetylation rate. The determining factor may be the inhibition of active species in the course of the reaction [90].

## 3. On the Mechanism of the Chitin Deacetylation Reaction

It is advisable to divide the discussion of the mechanism of the chitin deacetylation reaction into two parts, considering separately the mechanism for the first and second segments of the kinetic curve.

### 3.1. The Reaction Mechanism in the First Section of the Kinetic Curve

When considering the mechanism that determines the course of the reaction in the first section of the kinetic curve, special attention is paid to the effect of the hydration (solvation) of particles, the formation of an active particle that causes deacetylation, and the influence of the nature and concentration of alkali on the reaction rate.

#### 3.1.1. Hydration of Chitin/Chitosan and Sodium Hydroxide

In our opinion, the most probable factor affecting the kinetics of the deacetylation reaction is the formation of hydrate (solvate) complexes as a result of the interaction of chitin/chitosan macromolecules and sodium hydroxide with a solvent (water). It was shown in [62] that the rate of deacetylation of wet (previously kept in water) chitin is significantly lower than the rate of deacetylation of the initial dry chitin (Figure 6). Apparently, the hydration of chitin macromolecules leads to a decrease in the reaction rate in the first part of the kinetic curve.

We tried to separate the hydration process and the deacetylation reaction, taking into account that the latter starts to proceed at a noticeable rate at a high temperature (about 100 °C.) For this, the initial chitin was kept in an alkali solution and in water at room temperature to form a hydration shell before deacetylation. Pre-treatment of chitin with 50% NaOH solution did not lead to a decrease in DD (DD = (59 ± 1)%), while treatment with water led to a decrease in DD after deacetylation to DD = (39 ± 3)% (Figure 10). This indicates the formation of a hydration shell of chitin macromolecules in water and the absence of such a shell in a 50% NaOH solution.

#### 3.1.2. Alkali Concentration and the Nature of the Active Particle

It has been reported in many studies that the rate of the deacetylation reaction increases with increasing NaOH concentration [56,72,91], but not monotonously (Figure 11a, curve 1). The degree of deacetylation sharply increases starting from a NaOH concentration of 20–25 wt.%. In this concentration range, there is also a maximum specific electrical conductivity (s) of sodium hydroxide solution (Figure 11a, curve 2). These experimental data can be explained by the complete hydration of sodium hydroxide ions, in which water molecules bind into stable hydrate complexes of cations and anions, leading to a decrease in the number of “free” water molecules. In [92], the existence of a limit of complete hydration (LCH) was proposed, at which all water molecules are associated with ions and molecules of dissolved substances (Figure 11b), being part of the primary hydration shell. We assumed that the deacetylation reaction starts when passing through LCH, i.e., under conditions when the number of water molecules is no longer sufficient to form complete hydration shells.

In accordance with the reaction mechanism of alkaline deacetylation of the acetamide bond, the role of nucleophilic particles is played by hydroxide ions [66,93], whose reactivity increases with a decrease in their degree of hydration. Apparently, chitin can also form a hydrated complex with water, while the hydrated chitin molecule participates in the deacetylation reaction at a slower rate.

With an increase in alkali concentration, the reactivity of hydroxide ions increases (Figure 11) as a result of a decrease in their degree of hydration due to a decrease in water concentration [69,92].

#### 3.1.3. The Nature of Alkali and the Number of Hydroxide Ions Hydration

The effect of hydration of alkali ions on the kinetics of the deacetylation reaction is confirmed by experiments (Figure 12) using NaOH and KOH solutions. The first assumption is that in the case of different cations (Na^+^ or K^+^) with different ion diameters (d_i_) and different hydration energies (G_h_), the concentration of “free” water in the system will differ. The hydration energy and size of the K^+^ ion are G_h_ = 398 kJ/mol and d_i_ = 0.102 nm; for Na^+^, G_h_ = 271 kJ/mol and d_i_ = 0.138 nm [94,95]. This difference should affect the degree of hydration of hydroxide ions competing with cations for water molecules in the hydration shell and, consequently, the concentration of “free” water, the reaction rate, and the shape of the chitin deacetylation kinetic curve.

The kinetic curves of deacetylation in different alkalis at the same molar concentration of alkali (13.48 mol/dm^3^) differ from each other (Figure 12, curves 1 and 2). Contrary to the assumption, a higher rate in the first part of the curves is observed in a KOH solution with a lower hydration energy of the K^+^ ion. This is probably due to different molar concentrations of water: 48.49 mol/dm^3^ in NaOH solution and 41.94 mol/dm^3^ in KOH solution. The kinetic curve in the NaOH solution with a higher alkali concentration, but the same water concentration as the KOH solution (41.94 mol/dm^3^) is shown in Figure 12 (curve 3). In this case, the rate of deacetylation in the first section was higher, which is due to a higher concentration of hydroxide ions (19.34 mol/dm^3^). However, at the same time, in contrast to the KOH solution, the reaction rate in the second section falls faster, and the “plateau” passes lower than in the KOH solution. The presented data indicate that with an increase in the concentration of alkali (hydroxide ions), the rate of the deacetylation reaction in the first section increases [43,65,69,72], and in the second it decreases [44].

A comparison of the kinetic curves of chitin deacetylation in NaOH and KOH solutions suggests that the state of the solution has a decisive influence on the deacetylation reaction. Apparently, the reactivity of hydroxide ions increases with an increase in the concentration of alkali as a result of a decrease in the degree of their hydration due to a decrease in the concentration of water. Thus, two factors act under these conditions: the first one is that a decrease in “free” water with an increase in alkali concentration reduces the degree of hydration of chitin, and the second one is that a decrease in the hydration of hydroxide ions increases their reactivity.

To elucidate the mechanism of the deacetylation reaction in the first section of the kinetic curve, we expressed the alkali content not as a percentage or molar concentration, but as the ratio of water molecules to one ion of the dissolved electrolyte. Knowing the molar concentration of alkali (C_al_) and water (C_w_), it is possible to calculate the number of water molecules per electrolyte ion C_w_/C_al_ (Table 1). In this case, almost identical dependences of the deacetylation reaction rate on C_w_/C_al_ are obtained (Figure 13).

In accordance with the calculations, it turned out that at an alkali concentration of 50%, there are less than 2 water molecules per electrolyte ion, therefore, the hydration shell of the ion is less than half filled (Table 1). Until now, the theory of highly concentrated electrolyte solutions seems to be empirical; in contrast to dilute solutions, the mathematical apparatus has practically not been developed. Thus, the authors of [97,98] believe that when passing through the limit of complete hydration (LCH), electrolyte ions begin to lose water molecules from hydration shells, which leads to a sharp change in the properties of the solution. At this moment, reactions begin that lead to a decrease in electrical conductivity, in particular, the formation of ion pairs and associates of oppositely charged ions.

According to our data, the deacetylation reaction begins to proceed at a noticeable rate when the number of water molecules per electrolyte ion (OH^−^) is less than that required for complete hydration—less than 6. Before LCH, OH^−^ ions have complete hydration shells and are practically inactive. Upon crossing the LCH, the hydration shell is depleted, and the chitin deacetylation reaction begins. Thus, the reactivity of hydroxide ions appears at the LCH point and further increases with a decrease in the degree of hydration (Figure 11 and Figure 13).

When discussing the mechanism of deacetylation, one should also take into account the fact that the number of ion pairs and associates of alkali molecules increases in concentrated solutions [93,99], which leads to a decrease in the concentration of “free” hydroxide ions. Thus, the actual concentration of hydroxide ions after the LCH point decreases. Consequently, the increase in the activity of hydroxide ions turns out to be greater than that observed in experiments. A similar conclusion was made in a monograph [100], where it was shown that the dependence of the activity of hydroxide ions on the water concentration is approximately 1/[H_2_O]^2^. The graph of the dependence of the activity of OH^−^ ions on the molar concentration of water is similar to the dependence obtained by us (Figure 13). Thus, water does not participate in the chitin deacetylation reaction but, on the contrary, inhibits the reaction.

### 3.2. Reaction Mechanism in the Second Section of the Kinetic Curve: Inhibition of the Deacetylation Reaction

The rate of the chitin deacetylation reaction in the second section decreases as shown in many studies [63,72,101,102,103], and the kinetic curve runs almost parallel to the abscissa axis (Figure 3). As the alkali concentration increases, the reaction rate decreases (Figure 12). As it was determined in [43], the activation energy of this step is about 48.76 kJ/mol as it.

It can be assumed that the deceleration of the deacetylation reaction is explained by the formation of a certain amount of water during the reaction. We hypothesized that this water, which is formed in close proximity to the chitin molecule, leads to chitin hydration, reducing the rate of further deacetylation. To determine the source of “free” water during deacetylation, consider the mechanism of the reaction of deacetylation.

The reaction of deacetylation—the reaction of alkaline hydrolysis of acetamide bonds (deacetylation reaction) is a bimolecular second-order nucleophilic substitution of SN_2_, in which a strong hydroxyl ion nucleophile attacks the acetamide bond [94,104]. As a result of the overall deacetylation reaction, no water is produced, and none is consumed. The detailed description of the mechanism includes several steps. In alkali solutions, when the amide carbon is attacked by a hydroxide ion (it can be conventionally called the “first hydroxide ion”), an anionic tetrahedral intermediate (T_O−_) is formed [105,106,107,108]. The reaction is reversible; therefore, the resulting intermediate can either turn into the starting compound or decompose into hydrolysis products. Decomposition pathways may include interaction with various forms in solution that transfer a proton from the hydroxide ion OH^−^ to the amide ion NHR^−^. In dilute solutions, water is postulated as a proton carrier [107].

At a high alkali concentration, the H^+^ proton is taken away by the “second” hydroxide ion OH^−^ to form a water molecule, and the second intermediate (T_O2−_) is formed, which then decomposes to form an acetate ion and an amide ion. The amide ion NHR^−^, obtained in this way, takes away the H^+^ proton from the water molecule [106]. The described mechanism of cleavage of the acetamide bond in a concentrated alkali solution can be depicted by the scheme in Figure 14, where R is the residue of the chitin/chitosan molecule.

It can be assumed [44] that in a concentrated solution of alkali, in which there is no “free” water, the amide ion NHR^−^, being a stronger nucleophile than the hydroxide ion OH^−^, takes away H+ from the water molecule included in the hydration shell of the hydroxide ion. In this case, the fact of an increase in the deacetylation rate with an increase in the concentration of alkali and a decrease in the concentration of “free” water speaks of the non-participation of water in the mechanism.

To explain the decrease in the deacetylation rate in the second section of the kinetic curve, we proposed two hypotheses: (1) the formation of “free” water and (2) the formation of a quasi-stable chitosan anion as a result of the reaction.

The formation of “free” water as a result of the deacetylation reaction: The hydration of chitin molecules

We assumed [44] that “free” water for the hydration of chitin is released from the hydration shell of the hydroxide ion when it is converted into an acetate ion. The hydration energy of the acetate ion is less than the hydroxide ion (Table 2), so the acetate ion is less hydrated than the hydroxide ion.

As a result, a certain amount of water is released, which locally leads to the hydration of the chitin macromolecule. Our experiments showed (Figure 9) that the rate of chitin deacetylation decreases with its hydration. In a local area near the surface of chitin particles, water and an acetate ion are formed, and hydroxide ions with sodium cations remain present in the solution (Figure 15). Thus, in accordance with the proposed scheme (Figure 15), the reaction can stop in the second part of the kinetic curve as a result of the formation of “free” water, which creates a hydration shell of the chitin molecule and slows down deacetylation.

The formation of “free” water can also affect the reactivity of hydroxide ions. It was found above that with an increase in the degree of hydration of hydroxide ions, the rate of deacetylation decreases. However, this contribution to the decrease in the deacetylation rate in the second section of the kinetic curve is probably less significant since the concentration of hydroxide ions in the solution is high and the overall change in the degree of hydration of hydroxide ions will be insignificant.

The proposed hypothesis does not explain the decrease in the deacetylation rate with an increase in the alkali concentration in the second section of the kinetic curve. It can be assumed that in a more concentrated alkali solution, the reactivity of hydroxide ions is higher due to a lower degree of hydration, but the concentration of hydroxide ions is lower due to the association of ions. Conversely, in a dilute solution, the concentration of hydroxide ions is higher, although their reactivity is lower. Apparently, this is precisely why in a more concentrated alkali solution the rate of deacetylation in the second section of the kinetic curve may be lower than in a more dilute solution. This speculative assumption requires a detailed study and accurate calculations of the state of hydroxide ions depending on the concentration of alkali.

#### Formation of Quasi-Stable Chitosan Amide Anion

The second hypothesis, which explains the deceleration of the deacetylation reaction in the second part of the kinetic curve, suggests the formation of a quasi-stable chitosan anion. It results from the cleavage mechanism of the acetamide bond (Figure 14).

The amide anion of chitosan NHR^−^ is likely to be stabilized in a concentrated alkali solution. In this case, an excess negative charge accumulates on the deacetylated chitin units, which will electrostatically repel hydroxide ions and reduce the deacetylation rate. If this anion is sufficiently stable, then the deacetylation reaction may stop; this is evidenced by the almost horizontal appearance of the kinetic curve in the second section (Figure 3) at a high concentration of NaOH. At a lower concentration of alkali, the charge of the amide anion is neutralized as a result of the addition of a proton from a water molecule (as a result of hydrolysis). In this case, the negative charge on the chitin molecule will decrease, and the rate of the deacetylation reaction will increase. These transformations can be represented by the following scheme (Figure 16). As a result of the reaction, we see the formation of an amide anion of chitosan (Figure 16a), which can be neutralized upon interaction with a water molecule (Figure 16b).

At a high alkali concentration, “free” water is practically absent, and the chitosan anion “borrows” a water molecule from hydrated electrolyte ions. With an increase in the concentration of alkali, that is, with a decrease in the degree of hydration of alkali ions, the binding energy of alkali ions with water molecules increases, therefore, the probability of detachment of a water molecule by the chitosan anion decreases. Thus, the lifetime of the chitosan anion will increase. Such a quasi-stable anion creates an electrostatic obstacle for the hydroxide ion to attack chitin molecules, which acquire a negative charge. As a result, the rate of the deacetylation reaction decreases, and the more it decreases, the higher the alkali concentration.

The quasi-stable amide anion of chitosan is destroyed by water. Therefore, after washing the intermediate, the deacetylation starts again. Since the hydration of chitin slows down the reaction rate in the first site (Figure 9), in order to rapidly deacetylate, it is necessary to dry the washed intermediate.

Chitin molecules during the deacetylation reaction under heterogeneous conditions are in an ordered state, as a result of the formation of hydrogen bonds between them [111,112]. Under such conditions, the negative charge on the amide group of one polysaccharide molecule can interact with the carbon atom in the acetamide group of the neighbouring molecule, on which there is a partial positive charge (Figure 17).

As a result, the value of the negative charge on the nitrogen atom of the amide group of one molecule decreases, and the positive charge on the carbon atom of the acetamide group of the second molecule is partially neutralized. In this case, the nucleophilicity of the amide group of the first molecule decreases (which makes an additional contribution to the stabilization of the amide ion) and the electrophilicity of the carbon atom of the acetamide group of the second molecule decreases. As a result, the probability of detachment of a proton by an amide ion from a water molecule decreases, and the rate of the nucleophilic reaction decreases. In general, this can lead to a complete stop of the deacetylation reaction. With an excess of water molecules (for example, during washings), the amide ion passes into a neutral form (Figure 16b), and its effect on the acetamide group of the neighbouring molecule disappears.

Figure 18 shows the effect of water was demonstrated by its addition to the reaction mixture after the kinetic curve reached the second region. Additional water causes the disappearance of the chitosan anion, which allows further deacetylation [44].

## 4. Conclusions

This review considers, in detail, the kinetic features of the reaction of heterogeneous deacetylation of chitin in concentrated (50 wt.%) alkali solutions and analyzes the reasons for the inhibition of the reaction, which make it impossible to obtain chitosan with a limiting degree of deacetylation of 100% as a product of this reaction. The dependence of the degree of deacetylation on the reaction time has a characteristic shape consisting of two sections. The first site (or site of fast deacetylation) is characterized by a sharp increase in the degree of deacetylation within about 30 min from the start of the reaction. The second site (or the site of slow deacetylation) is characterized by a slow increase in degree of deacetylation due to a sharp decrease in the reaction rate. After 3–4 h of reaction time, the maximum achievable value for the degree of chitosan deacetylation, equal to about (80–85)%, is reached.

A mechanism for the reaction of chitin deacetylation is proposed, which explains the features of the reaction kinetics. An analysis of publications on the decrease in the rate of the deacetylation reaction was carried out, and some new experimental results were presented, confirming the conclusions of the authors. It has been shown that water present in the reaction mixture has the greatest effect on the kinetics of chitin deacetylation, and it can lead to hydration of both chitin/chitosan macromolecules and alkali ions. Within the framework of the mechanism under consideration, a hypothesis is proposed according to which there is a dynamic equilibrium in the reaction mixture between hydrated alkali ions, chitin molecules, and the resulting acetate ions, which can shift depending on the concentration of the reacting particles.

It has been suggested that in highly concentrated aqueous solutions of alkalis, water is almost completely in the form of hydration shells of alkali ions and, to a lesser extent, in the form of “free” water. Apparently, the rate-limiting step in the deacetylation reaction is the hydration of chitin macromolecules by water molecules, that are released during the nucleophilic substitution of acetyl radicals associated with the amino groups of chitins by hydroxyl ions. The acetate ion is the product of the deacetylation reaction. The hydration energy of the acetate ion is less than the hydration energy of the hydroxide ion, so the acetate ion is less hydrated, resulting in some “free” water being released. Under these conditions, hydration of chitin macromolecules occurs in a local region around chitin molecules, where a local high concentration of water is created after deacetylation. The concentration of alkali in the entire reaction volume remains almost constant.

Establishing the mechanism of the chitin deacetylation reaction is of practical importance for the development of technologies for the production of chitosan, and it also makes a theoretical contribution to the understanding of the role of the solvent and hydration processes when considering the kinetics of heterogeneous reactions.

## Figures and Tables

**Figure 1 polymers-15-01729-f001:**
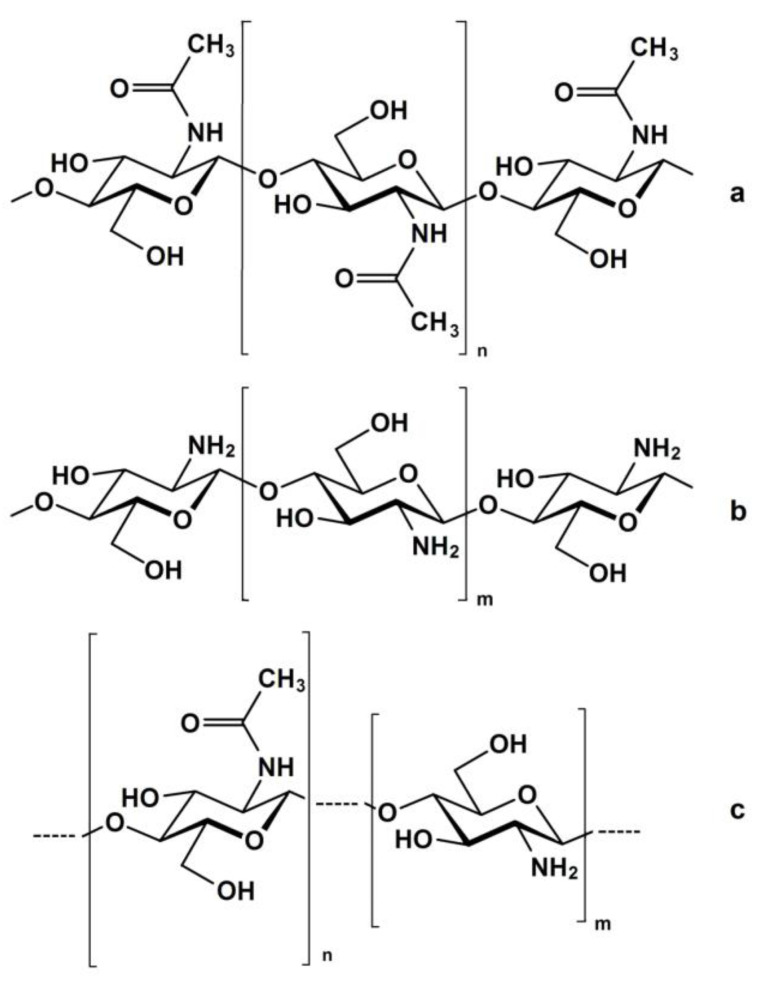
Chemical structure of (**a**) completely acetylated chitin [polyβ-(1→4)-N-acetyl-D-glucosamine], (**b**) completely deacetylated chitosan [polyβ-(1→4)-D-glucosamine], and (**c**) commercial chitin or chitosan, a copolymer characterized by its average degree of deacetylation (DD, %).

**Figure 2 polymers-15-01729-f002:**
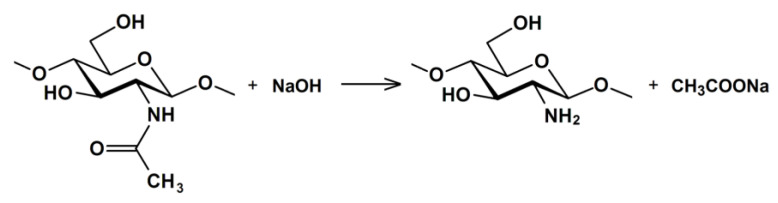
Reaction scheme for N-deacetylation of chitin.

**Figure 3 polymers-15-01729-f003:**
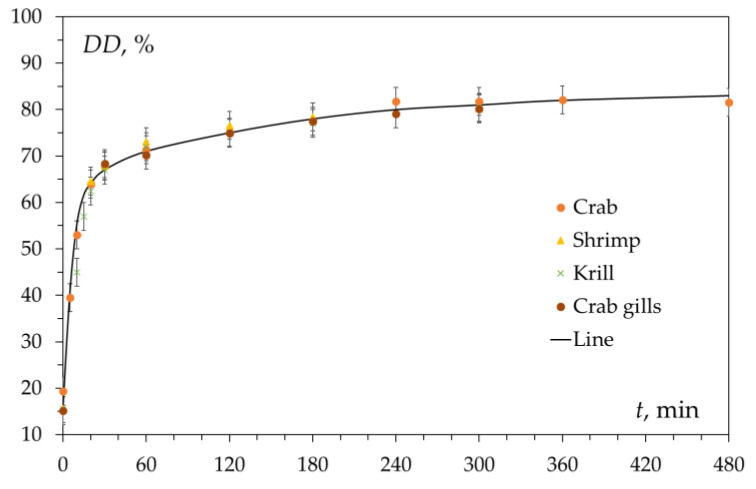
Kinetic curves of deacetylation of chitin obtained from crab shell, shrimp, Antarctic krill, and red king crab gills. Reaction conditions: 50% NaOH; 95 ± 2 °C; DD_0_ 15–20%. Original figure.

**Figure 4 polymers-15-01729-f004:**
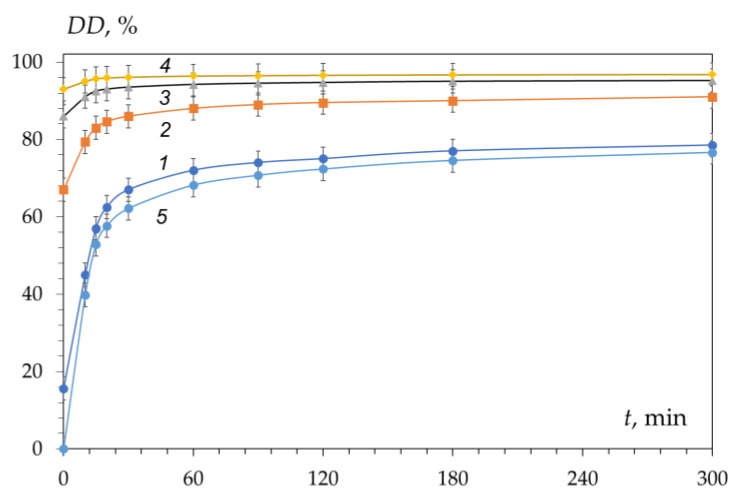
Kinetic curves for the deacetylation of chitin/chitosan with different DD_0_: 15.6% (1), 67% (2), 86% (3), 93% (4), and the “standard” curve (5). Reaction conditions: 50% NaOH; 95 ± 2 °C. Original figure.

**Figure 5 polymers-15-01729-f005:**
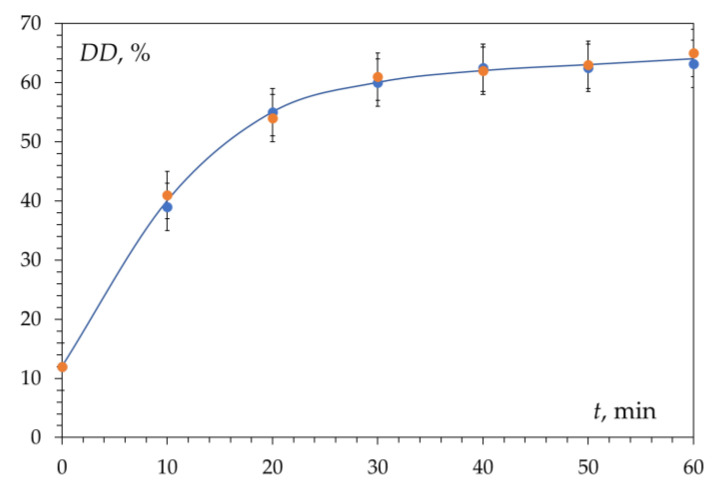
Kinetic curves of chitin deacetylation in 50% NaOH (●) and in 50% NaOH with the addition of 5% CH3COONa (●). Original figure.

**Figure 6 polymers-15-01729-f006:**
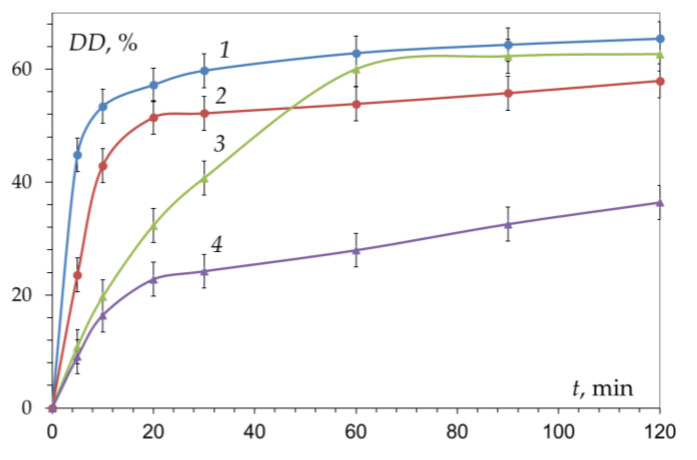
Kinetic curves of chitin deacetylation: 1—initial dry; 2—initial wet; 3—dry reprecipitated from solution in HCl; and 4—wet reprecipitated from HCl solution. Conditions: 50% NaOH, 100 °C. Original figure.

**Figure 7 polymers-15-01729-f007:**
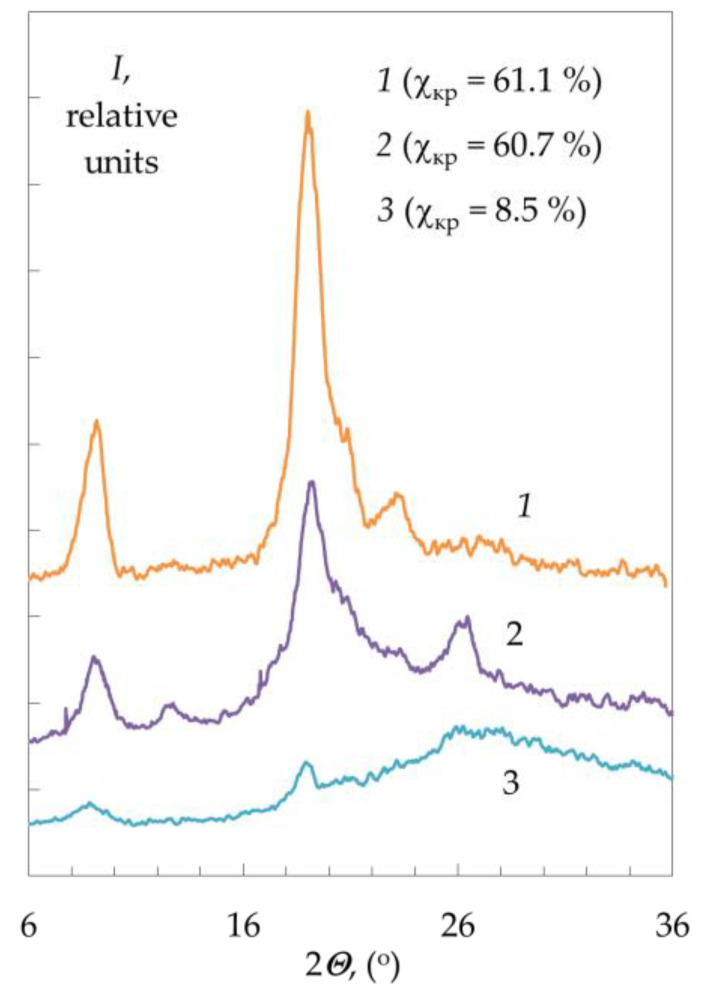
Diffraction patterns of samples of initial chitin (1) and chitin reprecipitated from HCl, dried in air (2), and wet (3) with different degrees of crystallinity (indicated in the figure). Original figure.

**Figure 8 polymers-15-01729-f008:**
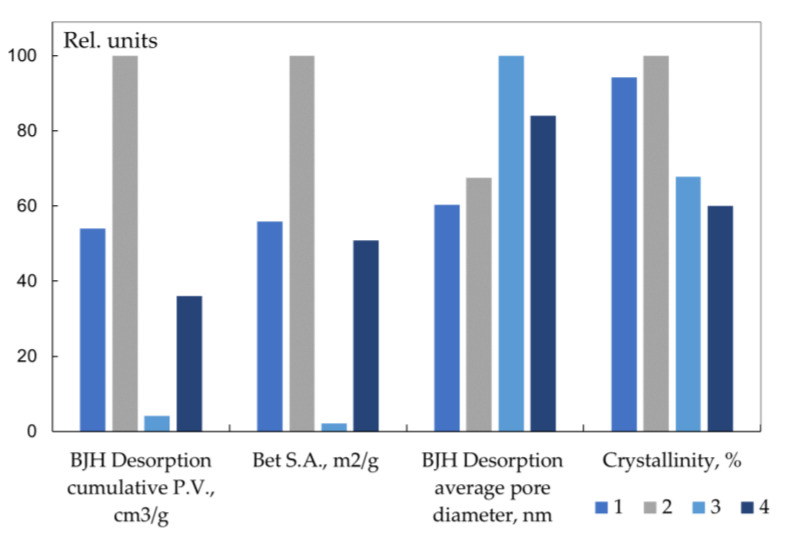
Indicators of pore volume, specific surface area, pore diameter, and crystallinity of chitin samples: original chitin dried in air (1) and in a freeze dryer (2); 3 and 4—reprecipitated chitin dried in air (3) and after freeze-drying (4). Original figure.

**Figure 9 polymers-15-01729-f009:**
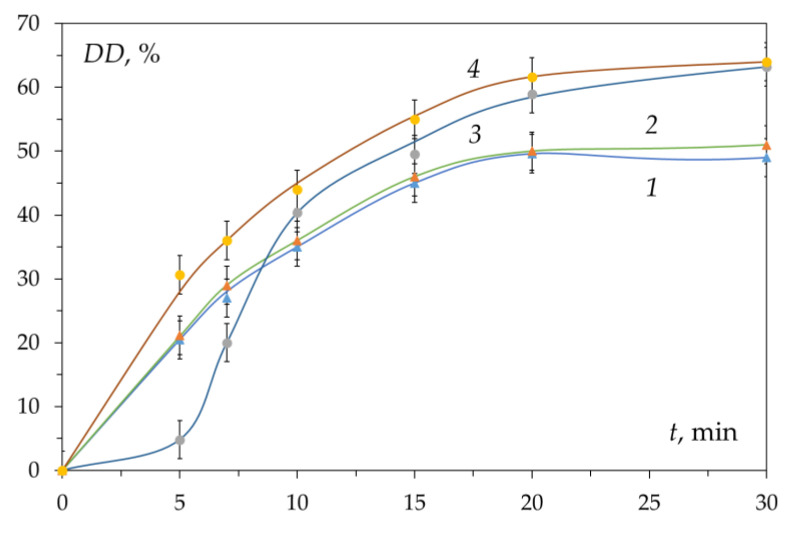
Kinetic curves of deacetylation of chitin samples in 50% NaOH at (95 ± 1) °C: initial chitin dried in air (1) and in a freeze dryer (2); reprecipitated chitin dried in air (3) and in a freeze dryer (4). Original figure.

**Figure 10 polymers-15-01729-f010:**
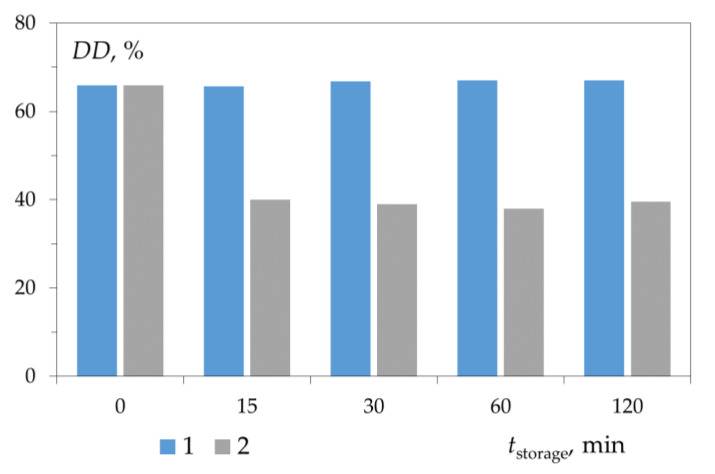
Dependence of the degree of deacetylation on the duration of preliminary exposure of chitin in 50% NaOH solution (1) and in water (2) at 20 °C. Deacetylation: T = 100 °C, τ = 30 min, DD0 = 18.7%. Original figure.

**Figure 11 polymers-15-01729-f011:**
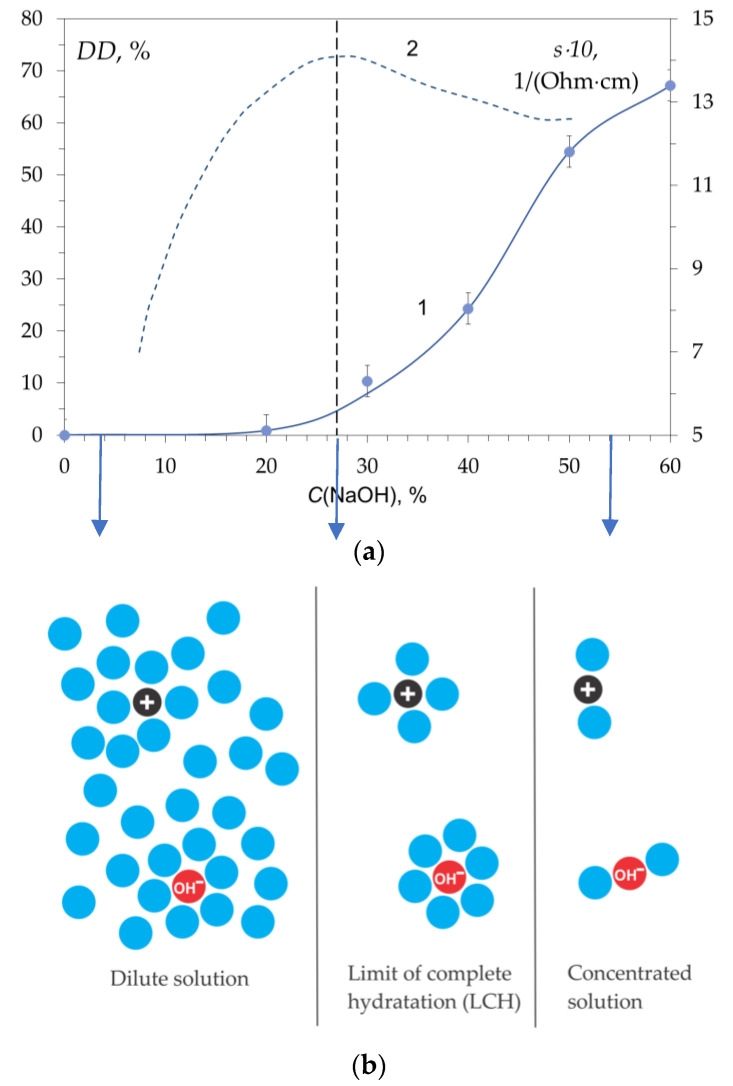
(**a**) Deacetylation degree, DD, (1) and electrical conductivity of the NaOH solution, s, (2) as a function of alkali concentration at T = 100 °C. (**b**) Scheme describing the formation of existence of a limit of complete hydration (LCH). Original figure.

**Figure 12 polymers-15-01729-f012:**
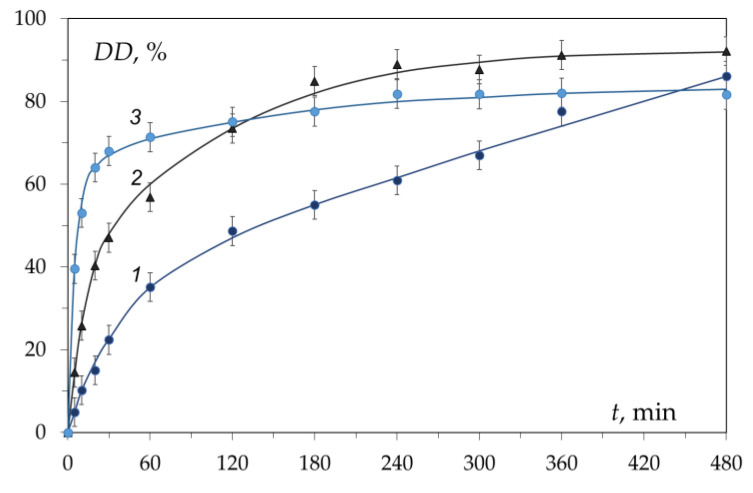
Kinetic curves of heterogeneous alkaline deacetylation at 100 °C. Solution concentrations (mol/dm^3^): NaOH, 13.48 (1), KOH, 13.48 (2), and NaOH, 19.34 (3). The mass ratio of chitin and solution is 1:55. Original figure.

**Figure 13 polymers-15-01729-f013:**
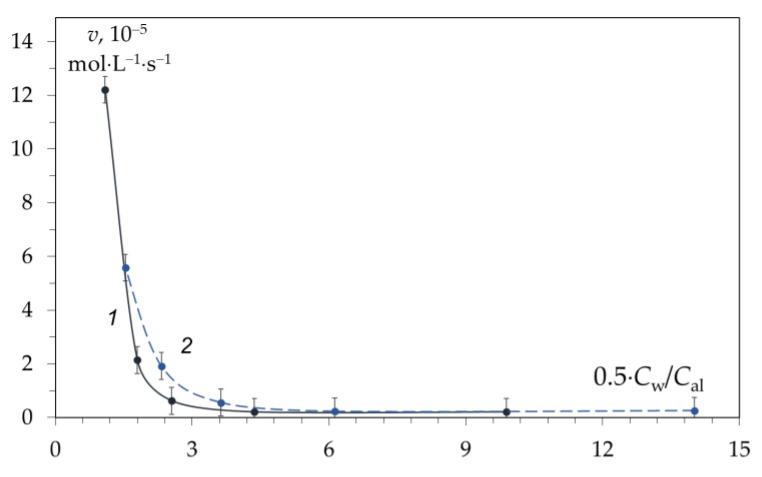
Dependence of the rate of the deacetylation reaction in the first section of the kinetic curve during the deacetylation time of 0–10 min at 100 °C in a solution of NaOH (1) and KOH (2) on the number of water molecules per one ion of dissolved hydroxide. Original figure.

**Figure 14 polymers-15-01729-f014:**
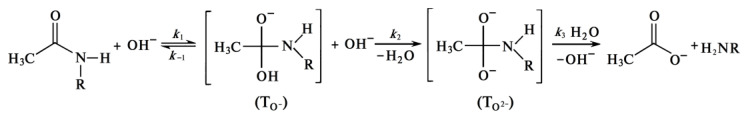
Scheme of the mechanism of cleavage of the acetamide bond in a concentrated alkali solution.

**Figure 15 polymers-15-01729-f015:**
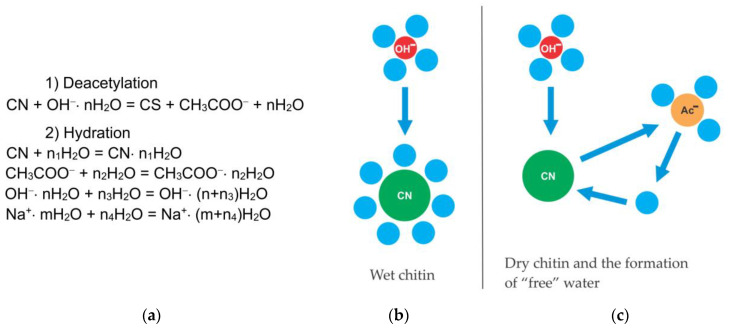
Competitive mechanism of hydration of reaction particles in the local region of the deacetylation reaction. (**a**) Reactions; (**b**), (**c**)—scheme of hydration of reaction particles. CN—chitin; CS—chitosan; Ac^−^—acetate ion; OH^−^—hydroxide ion. Original figure.

**Figure 16 polymers-15-01729-f016:**
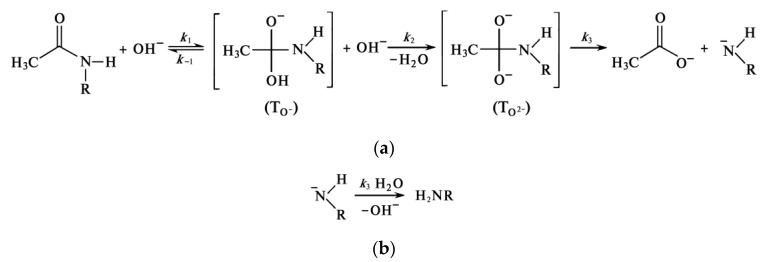
Scheme of (**a**) formation of the amide anion of chitosan NHR- and (**b**) its hydrolysis.

**Figure 17 polymers-15-01729-f017:**
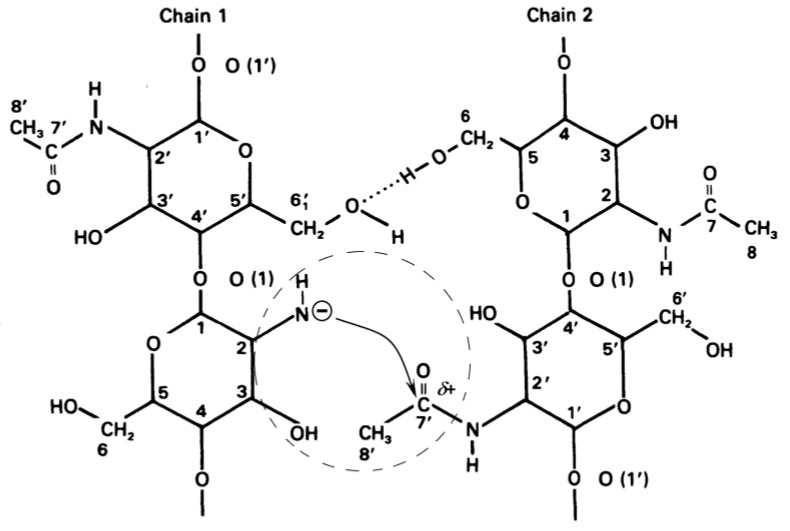
Scheme of the influence of the negative charge of the quasi-stable amide anion of chitosan on the electron density of the carbon of the acetamide bond of the neighboring chitin molecule.

**Figure 18 polymers-15-01729-f018:**
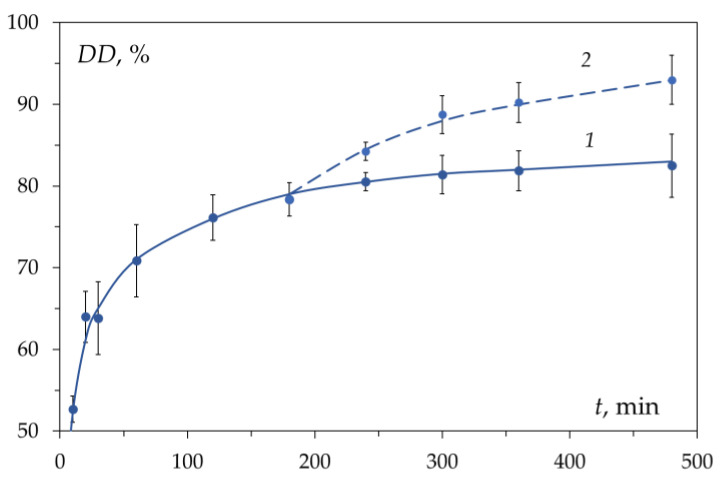
The kinetic curve of chitin deacetylation in 50 wt.% NaOH solution at T = 100 °C without added water (1) and with the addition of water (2) 3 h after the start of the reaction to a NaOH concentration of 40 wt.%. Original figure.

**Table 1 polymers-15-01729-t001:** Concentrations of NaOH and KOH solutions, calculated values of the number of water molecules per hydroxide ion (0.5 × C_w_/C_al_), and hydration number (HN) of ions.

Characteristics	KOH	NaOH	NaOH	LCH	
				KOH	NaOH
ω_alkali_, %	50	38	50	23.8	18.2
C_al_, mol/dm^−3^	13.5	13.5	19.4	5.18	5.45
C_w_, mol/dm^−3^	41.9	48.5	41.9	51.84	54.48
C_w_/C_al_	3.1	3.6	2.2	10	10
0.5 × C_w_/C_al_	1.55	1.80	1.10	5	5
HN for M^+^	data			4 *	4 *
HN for OH^−^				6 *	6 *

* Values are taken from [96].

**Table 2 polymers-15-01729-t002:** Hydration energy of ions.

Ions	ΔG_h_, kJ/mol	
Hydroxide ion OH^−^	444 [109]	430 [110]
Sodium cation Na^+^	398 [94]	365 [110]
Acetate ion CH_3_COO^−^	322 [94]	365 [110]

## Data Availability

Not applicable.

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
