# Peer review of "Mechanism of Heterogeneous Alkaline Deacetylation of Chitin: A Review"

_polymers, 2023, doi:10.3390/polym15071729_

Round 1
Reviewer 1 Report
The Review on "Mechanism of Heterogeneous Alkaline Deacetylation of Chitin:" has been explicitly written. However the percentage of plagiarism is quite high and thus the manuscript should take care of self plagiarism as well.
Author Response
Response to Reviewer 1 Comments
The authors thank the highly respected expert for his attention to the manuscript and the comments made. Below are answers to questions and comments.
Comments: The Review on "Mechanism of Heterogeneous Alkaline Deacetylation of Chitin:" has been explicitly written. However, the percentage of plagiarism is quite high and thus the manuscript should take care of self-plagiarism as well.
Response 1: In this review, we tried to analyze the main results describing the reaction of chitin deacetylation in an alkaline medium, presented in modern literature. In the review, the authors also presented their results on this topic, which were obtained over the past 10 years, and analyzed them taking into account the literature data.
To reduce the share of self-citation, we edited (reduced) part of the text, and inserted two new figures/schemes (Figure 11b, Figure 15b,c) taking into account the wishes of respected reviewers.

Reviewer 2 Report
In this review, authors systematically summarized experimental data of heterogeneous alkaline deacetylation of Chitin from authors' work and other groups. They proposed the reasonable mechanism of the deacetylation reaction. It is meaningful to both fundamental studies and also practical applications. However, several questions need to be solved to make the manuscript acceptable.
Question 1: Mistakes and typos need to be corrected. Such as in lines 308 and 314 "α- and -chitins", should be two types of chitins.
Question 2: The sentence in lines 169-170, "A sharp decrease in the reaction rate in the second 169 section does not make it possible to obtain chitosan with DD = 100%." seems not logic. Why the sharp decrease makes it impossible to reach DD=100%? Although the rate is very slow, it still increases. In principle, it would be 100% if the reaction run very long time. Authors should rephrase the language to make it clearer.
Question 3: Figure 5, the capture and legend label don't match the figure. black dots and white dots in capture but blue dots and yellow dots in figure.
Question 4: Figure 11, the capture is not correct.
Question 5: When citing a reference, it's be better to mention authors in that reference and what they did.
Question 6: In section 3, "Diffusion of hydroxide ions" and "porosity of chitin", it would be better to present a graphic scheme demonstrating how these factors affect the deacetylation reaction to make readers understand more easily.
Question &: authors mainly discussed their method to build up the mechanism, other methods should be compared or mentioned in introduction.
Author Response
Response to Reviewer 2 Comments
The authors thank the highly respected expert for his attention to the manuscript and the comments made. Below are answers to questions and comments.
Question 1: Mistakes and typos need to be corrected. Such as in lines 308 and 314 "α- and -chitins", should be two types of chitins.
Response 1: Thank you for the remark. The typos are corrected. In the lines 308 and 314 the second type of chitin b-chitin is indicated.
Question 2: The sentence in lines 169-170, "A sharp decrease in the reaction rate in the second 169 section does not make it possible to obtain chitosan with DD = 100%." seems not logic. Why the sharp decrease makes it impossible to reach DD=100%? Although the rate is very slow, it still increases. In principle, it would be 100% if the reaction run very long time. Authors should rephrase the language to make it clearer.
Response 2: I agree with the remark. Indeed, numerous experimental data show that with an unlimited increase in the reaction time, the degree of deacetylation does not reach 100%. We paraphrased the phrase (lines 169-171).
Question 3: Figure 5, the capture and legend label don't match the figure. black dots and white dots in capture but blue dots and yellow dots in figure.
Response 3: Thank you for your remark. I have corrected the legend in Figure 5. In the corrected version, the color of the marker in the legend matches the color of the marker on the curve.
Question 4: Figure 11, the capture is not correct.
Response 4: Thank you for your remark. It was error (typo). This is actually figure 12. I inserted a new caption under the picture on page 13 (line 441-443): “Figure 12. Kinetic curves of heterogeneous alkaline deacetylation at 100 oC. Solution concentrations (mol/dm3): NaOH, 13.48 (1), KOH, 13.48 (2), and NaOH, 19.34 (3). The mass ratio of chitin and solution is 1:55. Original figure.”
Question 5: When citing a reference, it's be better to mention authors in that reference and what they did.
Response 5: Thank you for your remark. However, I think that it is not necessary to write names of some authors in the text. Otherwise, the other authors should be mentioned. By this reason, it would enough sufficient to give the number of Reference (e.g. in Ref [9] or [59,65,72]).
Question 6: In section 3, "Diffusion of hydroxide ions" and "porosity of chitin", it would be better to present a graphic scheme demonstrating how these factors affect the deacetylation reaction to make readers understand more easily.
Response 6: Thank you for your offer. For easier reading and better understanding, I have made changes to the text of this section - part of the text has been shortened and paraphrased the main sentences. Changes are made to the text in red font.
Question 7: authors mainly discussed their method to build up the mechanism, other methods should be compared or mentioned in introduction.
Response 7: Thank you for your remark. We compared our results with kinetic models proposed in various papers, for example:
- Bradić, B.; Bajec, D.; Pohar, A.; Novak, U.; Likozar, B. A reaction–diffusion kinetic model for the heterogeneous N-deacetylation step in chitin material conversion to chitosan in catalytic alkaline solutions. React. Chem. Eng. 2018, 3, 920–929. DOI:10.1039/C8RE00170G
- Jiang, C.J.; Xu, M.Q. Kinetics of heterogeneous deacetylation of β-Chitin. Chem. Eng. Technol. 2006, 29(4), 511–516. DOI:10.1002/ceat.200500293
- de Souza, J.R.; Giudici, R. Effect of diffusional limitations on the kinetics of deacetylation of chitin/chitosan. Carbohydr. Polym. 2021, 254, 117278. DOI:10.1016/j.carbpol.2020.117278.
The experimental results obtained by the authors were compared and discussed with the kinetic and diffusion models of the deacetylation reaction – lines 171-172, 187-182, 295-301, 338-341, 408-409, 413-415, 437-439.

Reviewer 3 Report
This review is worthy of publication with the following edits:
1) Remove - in-front of Chitin (line 159)
2) Edit the caption for Figure 4
3) Line 245: Instead of addressing as a Japanese researcher, cite the name of the authors.
4) Draw a mechanism for lines 398-411 or any other visual representation
5) Add the description for 3 in Figure 11
Author Response
Response to Reviewer 3 Comments
The authors thank the highly respected expert for his attention to the manuscript and the comments and remarks made. Below are answers to questions and comments.
This review is worthy of publication with the following edits:
- Remove - in-front of Chitin (line 159)
Response 1: Thank you for the remark. In line 159, the typo has been corrected, it says "alpha-chitin".
- Edit the caption for Figure 4
Response 2: Figure 4 caption has been edited as follows: «Kinetic curves for the deacetylation of chitin/chitosan with different DD0: 15.6% (1), 67% (2), 86% (3), 93% (4) and the "standard" curve (5). Reaction conditions: 50% NaOH; 95 2 oC.»
- Line 245: Instead of addressing as a Japanese researcher, cite the name of the authors.
Response 3: Thank you for your remark. However, I think that it is not necessary to write names of some authors in the text. Otherwise, the other authors should be mentioned. By this reason, the proposal was edited as follows: “This hypothesis was put forward in [74] and continues to be used in various works [59,65,72]”. (Lines 246-247)
- Draw a mechanism for lines 398-411 or any other visual representation
Response 4: I agree with your proposal. Figure 11 has been modified, a scheme has been added explaining the formation of existence of a limit of complete hydration (LCH). In addition, Figure 15 has been modified and supplemented with a diagram that explains the formation of "free" water, which plays an important role in the reaction mechanism of alkaline deacetylation of chitin
- Add the description for 3 in Figure 11
Response 5: Scheme describing the formation of existence of a limit of complete hydration (LCH) has been added to the Figure 11. Description of the scheme is given in the manuscript (lines 394-414).
